# Altered Distribution of Unesterified Cholesterol among Lipoprotein Subfractions of Patients with Diabetes Mellitus Type 2

**DOI:** 10.3390/biom13030497

**Published:** 2023-03-08

**Authors:** Livia Noemi Kolb, Alaa Othman, Lucia Rohrer, Jan Krützfeldt, Arnold von Eckardstein

**Affiliations:** 1Institute of Clinical Chemistry, University of Zurich and University Hospital of Zurich, CH-8091 Zurich, Switzerland; 2Institute of Molecular Systems Biology, ETH Zurich, CH-8049 Zurich, Switzerland; 3Department of Endocrinology, Diabetology and Clinical Nutrition, University Hospital of Zurich, CH-8091 Zurich, Switzerland

**Keywords:** diabetes mellitus type 2, biomarker, lipoprotein subclasses, nuclear magnetic resonance spectroscopy, free cholesterol

## Abstract

Biomarkers are important tools to improve the early detection of patients at high risk for developing diabetes as well as the stratification of diabetic patients towards risks of complications. In addition to clinical variables, we analyzed 155 metabolic parameters in plasma samples of 51 healthy volunteers and 66 patients with diabetes using nuclear magnetic resonance (NMR) spectrometry. Upon elastic net analysis with lasso regression, we confirmed the independent associations of diabetes with branched-chain amino acids and lactate (both positive) as well as linoleic acid in plasma and HDL diameter (both inverse). In addition, we found the presence of diabetes independently associated with lower concentrations of free cholesterol in plasma but higher concentrations of free cholesterol in small HDL. Compared to plasmas of non-diabetic controls, plasmas of diabetic subjects contained lower absolute and relative concentrations of free cholesterol in all LDL and HDL subclasses except small HDL but higher absolute and relative concentrations of free cholesterol in all VLDL subclasses (except very small VLDL). These disbalances may reflect disturbances in the transfer of free cholesterol from VLDL to HDL during lipolysis and in the transfer of cell-derived cholesterol from small HDL via larger HDL to LDL.

## 1. Introduction

Diabetes mellitus type 2 (T2DM) is a leading health burden worldwide due to its high prevalence and high morbidity and mortality. Of the about 350 million patients diagnosed with diabetes in 2015, 80% lived in low-income or middle-income countries, with the highest proportion in Latin America [1]. Until 2035, the prevalence of T2DM is estimated to increase by 70% and 20% in developing and developed countries, respectively [1]. T2DM shortens life expectancy by about six years due to increased risks of atherosclerotic cardiovascular diseases (ASCVD), nephropathy, infections, and various cancers [2]. Moreover, retinopathy and neuropathy reduce the quality of life in many patients with T2DM. The prognosis of patients with diabetes mellitus can be much improved by lifestyle optimization as well as drug treatments that lower blood glucose, blood pressure, and LDL-cholesterol [3,4]. Nevertheless, diabetic complications still occur frequently and limit the quantity and quality of life in patients with diabetes [2]. To improve this situation, personalized treatments may be necessary. To this end, novel diagnostics and biomarkers are needed.

Nuclear magnetic resonance (NMR) spectroscopy allows the comprehensive quantification of metabolites in only 100 to 300 μL of plasma or serum [5]. By applying a high-frequency magnetic field, NMR spectroscopy records protons of molecules as specific resonance spectra. Specific algorithms allow the spectra to be assigned to specific molecules and quantified [6]. A particular strength of NMR spectroscopy is its ability to record different classes of lipids (cholesterol and cholesteryl esters, triglycerides, sphingomyelins, and glycerophospholipids) not only in total plasma but within specific lipoprotein subclasses without any prior fractionation of plasma, for example by ultracentrifugation. Fatty acids can be differentiated for the degree and position of denaturation. In addition, some metabolites such as amino acids and ketone bodies, as well as proteins such as apolipoproteins A-I and B and albumin and glycoprotein acetyls (Gp), are measured [7,8].

We applied this method to a case-control study to answer the following questions: Do biomarkers recorded by NMR spectroscopy of plasma differentiate diabetic subjects from non-diabetic controls? Do any of the NMR measures correlate with glycemic control as reflected by HbA1c?

## 2. Materials and Methods

### 2.1. Patients and Control Subjects

In total, 51 healthy volunteers and 66 patients with diabetes mellitus Type 2 were recruited at the Department of Endocrinology, Diabetology, and Clinical Nutrition and the Clinical Trial Center of the University Hospital Zurich [7]. All patients and controls attended with informed consent. The ‘Kantonale Ethikkommission Zürich’ granted the ethics approval (PB-BASEC_2015−00159). Exclusion criteria included acute illnesses or acute deterioration of a chronic illness, pregnancy, active inflammation (defined by leukocytes > 10 g/L or C-reactive protein > 10 mg/L), and advanced chronic kidney disease with an estimated glomerular filtration rate (eGFR) < 30 mL/min/1.73 m^2^. Table 1 describes the demographic and clinical features of the two patient groups.

### 2.2. Clinical Laboratory Tests

Glycohemoglobin (HbA1c) was measured with a fully automated High-Performance Liquid Chromatography system (ADAMS A1c from ARKRAY, Kyoto, Japan). Total cholesterol and HDL cholesterol, liver enzyme activities, C-reactive protein (CRP), creatinine, triglycerides, and glucose were quantified by the use of a cobas 8000 analyzer and assays from Roche diagnostics (Rotkreuz, Switzerland). LDL-cholesterol was computed with the Friedewald formula. ApoA-I and apoB were measured by immunonephelometry with the BN ProSpec analyzer and assays from Siemens Healthcare diagnostics (Erlangen, Germany). Lp(a) concentrations were measured by the application of a latex-enhanced immunoturbidimetric assay of Randox Laboratories Ltd. (Crumlin, UK) to a Konelab analyzer (Thermo Fisher, Waltham, MA, USA).

### 2.3. Nuclear Magnetic Resonance (NMR) Spectroscopy of Plasma

NMR spectroscopy of plasma was executed as a service for a fee by Nightingale Health (Helsinki, Finland: https://nightingalehealth.com/, accessed on 10 December 2022). The method records 154 variables, including cholesterol and cholesteryl esters, triglycerides, sphingomyelins, and glycerophospholipids in six subclasses of very low-density lipoproteins (VLDL), intermediate density lipoproteins (IDL), three subclasses of LDL, and four subclasses of HDL. It also allows quantifying numbers and average diameters of lipoproteins, as well as the quantitatively major apolipoproteins apoA-I and apoB. Fatty acids are differentiated into saturated (SAFA), monounsaturated (MUFA), and polyunsaturated fatty acids (PUFA). The latter are further differentiated for omega-3 (FAn3) and omega-6 fatty acids (FAn6). Among the omega-3 fatty acids, marine docosahexaenoic acid (DHA) and plant-derived linoleic acid (LA) are differentiated. Eight amino acids (alanine, glutamine, histidine, isoleucine, leucine, valine, phenylalanine, and tyrosine), three ketone bodies (acetate, acetoacetate, hydroxybutyrate), glucose, lactate, citrate, creatinine, albumin, and glycoprotein acetyls (Gp), a biomarker of inflammation, are also measured [7,8].

### 2.4. Statistics

The statistical evaluation of the data was performed with the software R [9]. Normality distribution was checked with the Shapiro–Wilk test. The parametric t-test and the non-parametric Wilcoxon test were applied for univariate pairwise comparisons of diabetic patients with control subjects for parameters with Gaussian (if Shapiro–Wilk test *p* < 0.05) and non-Gaussian frequency distribution (if Shapiro–Wilk test *p* ≥ 0.05), respectively. Correlations with HbA1c were calculated according to Spearman. *p*-values were adjusted for multiple testing according to the Benjamin–Hochberg method.

Volcano plots (fold change vs. *p*-value or coefficients of correlation vs. *p*-values) were generated using ggplot and ggrepel packages. For a clearer presentation, the variables were split into subgroups (clinical measures, VLDLs, LDLs, IDLs, HDLs, lipids, and non-lipids). ComplexHeatmap and Hmisc packages were used to construct heatmaps and do hierarchical clustering with the complete clustering method and the Spearman clustering distance.

Regularized logistic regression was performed to identify independent variables associated with T2DM. Regularized regression was essential to avoid over-fitting, considering that the dataset encompasses more variables than patients and because of the high degree of correlation between many variables. For that purpose, elastic net regularization was performed using the packages glmnet and c060. The elastic net regularization was performed as described before [7]. In short, concentrations below the level of quantification were imputed by the square root of the minimum value for each variable of the NMR data. Then, the data were log-transformed and scaled around the mean. To identify the best values for the regularization parameters, alpha and lambda values with the lowest root mean squared error (RMSE) were selected after tenfold cross-validation by logistic regression fitting. At alpha =1, the lowest RMSE was observed, so lasso regularization was performed. Lasso (least absolute shrinkage and selection operator) regularization shrinks the regression coefficient by minimizing the sum of their absolute values, thus forcing some coefficients into zero. That is performed to prevent overfitting and as a tool for variable selection. The non-zero regression coefficients after regularization were then calculated as the first output of the regularized logistic regression. Afterwards, stability path analysis was performed (as described earlier [7]). Stability path analysis compensates for some of the stochasticity of regularization in general, thereby providing more robust estimates of the association between independent variables and the T2DM state. Two stability path analyses were performed, with and without correction for multiple testing. The results from all analysis steps are reported.

## 3. Results

### 3.1. Univariate Associations with Diabetes

As expected, diabetic patients and healthy volunteers differ significantly in many anthropometric and clinical measures, as well as drug treatment (Table 1). Diabetic subjects were older, had higher BMI and waist circumference, higher heart rate, higher systolic and diastolic blood pressure, higher plasma levels of glucose, triglycerides, creatinine, and C-reactive protein (CRP), higher plasma activities of transaminases, higher HbA1c, hemoglobin, and hematocrit as well as higher blood counts for leukocytes and erythrocytes. Conversely, plasma concentrations of total HDL- and LDL-cholesterol, as well as apoA-I, were significantly lower in diabetic subjects.

Figure 1 shows a volcano plot of the associations of diabetes with these clinical measures as well as 154 features measured by NMR spectroscopy of plasma. All *p*-values are adjusted for multiple testing. Interestingly, several NMR-measured features are significantly different between patients with T2DM and non-diabetic control subjects—notably, the increase in branched-chain amino acids, alanine, and lactate in plasma samples of T2DM patients. In contrast, we observed a significant decrease in plasma levels of most features related to very large HDL (XL-HDL), large HDL (L-HDL), or related with polyunsaturated fatty acids (linoleic acid, omega-6 fatty acids) and monounsaturated fatty acids. In contrast to most plasma HDL-related parameters, those related to small HDL particles (S-HDL), especially triglycerides in small HDL (S-HDL-TG), were increased rather than decreased in T2DM patients. Similar opposite associations were seen for features related to very small VLDL (except XS-VLDL-TG), which are decreased in patients with diabetes by contrast to features related to small (S-VLDL), medium (M-VLDL), large (L-VLDL), very large VLDL (XL-VLDL), and chylomicrons and extremely large VLDL (XXL-VLDL), which are increased in T2DM patients. In contrast to PUFAs, plasma levels of monounsaturated fatty acids (MUFAs) were higher in diabetics. It is also noteworthy that the inverse association of apoA-I was stronger if it was measured by NMR spectroscopy (marked in red) rather than by the clinical laboratory method (marked in blue).

### 3.2. Univariate Correlations between Variables

Figure 2 and Figure 3 show heatmaps of correlations between all variables in non-diabetic control subjects and patients with diabetes, respectively. The clustering dendrograms on the top and left sides of the heatmaps reflect similarities between features in several clusters. The main findings are as follows:-In both control subjects and diabetic patients, clusters contain several features of one lipoprotein subclass or even several subclasses of one lipoprotein, for example, medium, large, and very large HDL or large, very large, and extremely large VLDL.-One noteworthy exception is triglycerides in small HDL (S-HDL-TG), which is separated from all other HDL parameters both in control subjects (cluster C2 vs. clusters C5 and C7 in Figure 2) and diabetic subjects (cluster D4 vs. cluster D3 in Figure 3).-Moreover, a cluster in control subjects (C4 in Figure 2) and diabetic subjects (D4 in Figure 3) is characterized by triglycerides rather than lipoprotein subclasses.-In the diabetes heatmap (Figure 3), some triglyceride-containing HDLs (M-HDL-TG, XL-HDL-TG, S-HDL-TG, HDL-TG), as well as S-LDL-TG and IDL-TG, are in the VLDL cluster D4.-The diabetes heatmap (Figure 3) contains triglyceride features in clusters otherwise characterized by cholesterol-rich lipoproteins (LDL-TG, M-LDL-TG, L-LDL-TG in cluster D1 and S-LDL-TG, IDL-TG, HDL-TG, XL-HDL-TG in cluster D4).-Features of small HDLs do not cluster with features of the other HDL (medium, large, or very large HDL). Two of them (S-HDL-PL and S-HDL-FC) in the diabetes heatmap (Figure 3) are even linked to the two LDL subclusters, D1 and D2.-VLDLs of different sizes are distributed among two subclusters (D4 and D5 in the diabetes heatmap shown in Figure 3, and C2 and C3 in the control heatmap shown in Figure 2). All features of very small VLDL (except XS-VLDL-TG, which clusters with other VLDL features) and some features of small VLDL (S-VLDL-C in diabetes, S-VLDL-CE in controls and diabetes) cluster with features of LDL in C1, respectively, in D1 and D2 rather than large and very large VLDL.

### 3.3. Multivariate Associations with Diabetes

Next, we tried to identify variables independently associated with the presence of diabetes. Considering the strong correlations between many variables and the fact that the dataset encompasses more variables than patients, we performed regularized logistic regression using elastic net regularization to avoid over-fitting (Figure 4 and Figure 5). After regularization, HbA1c, age, valine, lactate, glucose (measured either with the clinical method or NMR), free cholesterol in small HDL, leucine, isoleucine, waist circumference, and heart rate had positive non-zero regression coefficients (Figure 4A). In contrast, free cholesterol, linoleic acid, and total cholesterol measured by NMR and HDL-diameter had negative non-zero regression coefficients (Figure 4A).

Upon stability path analysis after elastic net regularization, only the associations of T2DM with HbA1c, age, lactate, and free cholesterol remained significant (Figure 4B). However, after correction for multiple testing, only HbA1c remained significantly associated with diabetes (Figure 4C).

### 3.4. Altered Distribution of Free Cholesterol among Lipoprotein Subclasses

The positive association of free cholesterol in small HDL particles with T2DM contrasts with the negative associations of free cholesterol in total plasma and most other HDL-related features with T2DM (Figure 1 and Figure 4A). We compared the associations of free cholesterol (FC), cholesterol ester (CE), and their ratio in all lipoprotein subclasses (Figure 5). The relative content of FC to CE in small LDL and both the absolute content of FC and the content relative to CE in small HDL as well as in all VLDL particles, except FC and CE in very small VLDL and CE in small VLDL, showed positive associations with T2DM. Conversely, both the absolute and relative FC content of medium, large, and very large HDL as well as large LDL and IDL, showed inverse associations with T2DM (Figure 5). Notably, the associations for absolute FC content were stronger than for relative FC content.

### 3.5. Correlations with HbA1c

Figure 6 shows a volcano plot of the correlations between HbA1c and all other variables in the combined cohort of diabetic patients and control subjects. As expected, the strongest correlations were seen for glucose. Other clinical features with significant positive correlations include waist circumference, triglycerides, age, leukocyte number, body weight, and BMI. HDL-cholesterol was inversely correlated. Interestingly, several NMR features have stronger correlations with HbA1c than these established risk factors. They include positive correlations of the branched-chain amino acids (BCAA: isoleucine, leucine, and valine), alanine, lactate, glycoprotein acetyls, and a series of VLDL features such as triglycerides in total, small, or medium VLDL, and VLDL-diameter. Many HDL-related features are inversely correlated with HbA1c, most strongly HDL-diameter. In agreement with this, features related to very large HDL had the strongest inverse correlations, followed by those of large and medium HDL. Several features related to small HDL instead showed positive correlations with HbA1c, namely (in the order of strength) triglycerides in small HDL, free cholesterol in small HDL, the number of small HDL particles, total lipids, and phospholipids in small HDL. In addition, several features related to polyunsaturated fatty acids also showed inverse correlations with HbA1c, namely omega-6 fatty acids, total polyunsaturated fatty acids, and linoleic acid.

Next, we performed elastic net regularized regression and stability path analyses to find variables independently correlated with HbA1c (Figure 7). Figure 7A shows the non-zero regression coefficients after regularization for the parameters with significant correlation. Glucose, leukocytes, age, alanine, and glycoprotein acetyls showed independent positive correlations. In contrast, the concentration of very large HDL particles, eGFR, and the linoleic acid to total fatty acids’ ratio showed independent negative correlations. Upon stability path analysis, only the correlations of HbA1c with glucose, leukocytes, age, alanine, and eGFR remained significant (Figure 7B). Upon correction for multiple testing, only glucose remained significantly correlated with HbA1c (Figure 7C).

## 4. Discussion

Upon maximal adjustment for confounders by elastic net analysis, our comprehensive analysis of clinical and metabolomics data of a case-control study with 117 subjects found diabetes and HbA1c associated and correlated, respectively, with a limited number of variables. We replicated well-established associations of diabetes with HbA1c, glucose, age, and waist circumference and likewise expected correlations of HbA1c with glucose, leukocytes, age, and estimated glomerular filtration rate. Our elastic net analysis also confirmed the independent associations of diabetes with several NMR-spectroscopic phenotypes, namely positively with branched-chain amino acids [10] and lactate [11], as well as inversely with linoleic acid in plasma [12] and HDL diameter [13,14]. HbA1c showed significant and independent correlations with alanine and acetylated glycoproteins (both positive), as well as linoleic acid relative to all fatty acids, and the number of very large HDL particles (inverse).

As a novel yet unreported observation, our elastic-net analysis with lasso regression found the presence of diabetes associated with lower concentrations of free cholesterol in plasma, but higher concentrations of free cholesterol in small HDL. Free cholesterol of small HDL was previously found to be associated with incident diabetes [15]. A closer look at all lipoprotein subfractions revealed decreased contents of free cholesterol in all HDL and LDL subclasses of diabetic subjects except small HDL, but higher concentrations of free cholesterol in all VLDL subclasses except in very small VLDL of diabetic subjects compared to non-diabetic controls. The larger the VLDL particle, the more pronounced the enrichment in free cholesterol.

In plasma, about 80% of cholesterol is esterified and transported in the core of lipoproteins. The unesterified cholesterol, frequently termed free cholesterol, is solved in the phospholipid layer of the lipoprotein surface. AcylCoA-cholesteryl-acyl transferase (ACAT) esterifies free cholesterol in the endoplasmic reticulum of enterocytes or hepatocytes, and the cholesteryl esters are secreted together with chylomicrons and VLDL, respectively [16]. The enrichment of VLDL in free cholesterol may be caused by reduced ACAT activity in diabetes. However, the data from animal experiments are controversial and showed decreased [17,18], increased [19,20,21], or unchanged ACAT activity [18,22] in the liver of diabetic rats or rabbits compared to control animals.

An alternative explanation for the enrichment of VLDL with free cholesterol is its impaired transfer to HDL. Kontush and colleagues previously demonstrated that the lipolysis of triglycerides is accompanied by the transfer of free cholesterol from triglyceride-rich lipoproteins to HDL, possibly as part of surface remnants [23,24]. This transfer is decreased in diabetes, probably as the result of disturbed lipolysis because insulin resistance limits the activity of lipoprotein lipase [23]. Such a disturbance explains both the accumulation of free cholesterol in VLDL and the decreased content of free cholesterol in HDL.

In plasma, lecithin:cholesterol acyltransferase (LCAT) esterifies free cholesterol, mostly in small HDL3, but also in larger HDL2 and even less in LDL [13,25]. The data on LCAT activity in diabetes is controversial. In agreement with the reduced content of free cholesterol in HDL and LDL, some studies reported increased LCAT activity in diabetic patients as compared to controls [26]. However, others found LCAT activity rather decreased or unchanged in diabetes [27]. The discrepancies may result from differences in the method used to determine LCAT activity. In the so-called exogenous substrate method, LCAT activity is determined by LCAT mass concentration, while in the endogenous substrate method, the composition of lipoproteins in plasma is strongly influencing the cholesterol esterification rate.

The opposite alterations in the free cholesterol content of small and larger HDL may reflect the flux and fate of cell-derived cholesterol within these lipoproteins. Small HDL is more effective than larger HDL in inducing cholesterol efflux from cells by the ATP-binding cassette transporter ABCA1. Only a part of cholesterol is immediately esterified in the initial and small HDL acceptor particles. Another part is transferred to larger HDL and then LDL without esterification [28,29,30]. The increased free cholesterol content of small HDL but decreased free cholesterol content of larger HDL and LDL may reflect a disturbed transfer of unesterified cholesterol between these lipoproteins. Interestingly, it was previously suggested that a high bioavailability of free cholesterol in HDL correlates with ASCVD [31]. Our observation of a reduced content of free cholesterol in most HDL particles of diabetic subjects, who are at high risk of ASCVD, rather contrasts this hypothesis. The hypothesis of Pownall and colleagues would not be falsified by our findings if the enrichment of free cholesterol in small HDL particles but deprivation in larger HDL and LDL indicates a block at this early stage of reverse cholesterol transport in diabetic subjects.

The major limitations of our study are the rather small population size, the monocentric design, and the lack of matching. This bears a high risk of confounding. We cannot rule out that our findings, although in line with previous reports, are not specific to diabetes but caused by differences, for example, in diet or drug treatment or ASCVD status, which were not included in our multivariate analysis, or body fat, as waist circumference also remained independently associated with diabetes (Figure 4A). The overrepresentation of males among diabetic subjects may have contributed to the findings, although sex was an independent determinant of neither diabetes nor HbA1c in our multivariate statistical analyses. Further studies are needed to replicate our novel finding on the association of diabetes with alterations in the content of plasma and lipoprotein subclasses in free cholesterol.

In conclusion, our comprehensive analysis of lipoproteins and metabolites using NMR spectroscopy replicated previously described associations of diabetes or correlations of HbA1c with alanine, branched-chain amino acids, lactate, polyunsaturated fatty acids, and sizes of lipoprotein subclasses. As a novel discovery, we found that most HDL particles, as well as LDL, are deprived of free cholesterol. Meanwhile, VLDL, as well as small HDL, are enriched with free cholesterol. This disbalanced distribution of free cholesterol may reflect disturbances in the transfer of free cholesterol from VLDL to HDL during lipolysis and in the transfer of cell-derived cholesterol from small HDL via larger HDL to LDL (see graphical abstract presented as Figure 8). In future research, it will be interesting and of potential clinical importance to unravel the molecular basis and pathogenic consequences of the disbalanced lipoprotein distribution of free cholesterol.

## Figures and Tables

**Figure 1 biomolecules-13-00497-f001:**
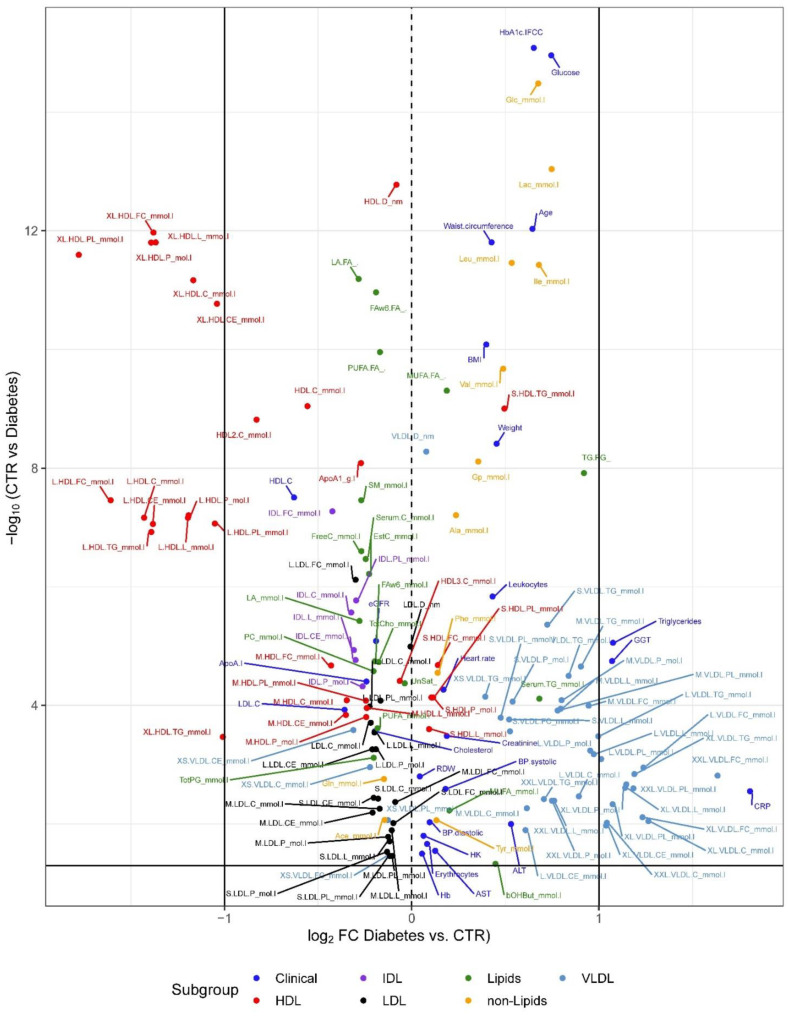
Volcano plot of associations between the presence of diabetes and clinical features or features analyzed by NMR spectroscopy of plasma. The x-axis reflects fold changes in the medians of the parameters; positive values signify an increase in diabetics, whereas negative values imply a decrease in diabetics. The y-axis shows the level of statistical significance. The higher the location of a parameter in the graph, the higher the statistical significance of its association with diabetes. The horizontal line reflects the threshold for statistical significance after being adjusted for 192 comparisons by the Benjamin–Hochberg procedure. For the list of abbreviations, see Appendix A.

**Figure 2 biomolecules-13-00497-f002:**
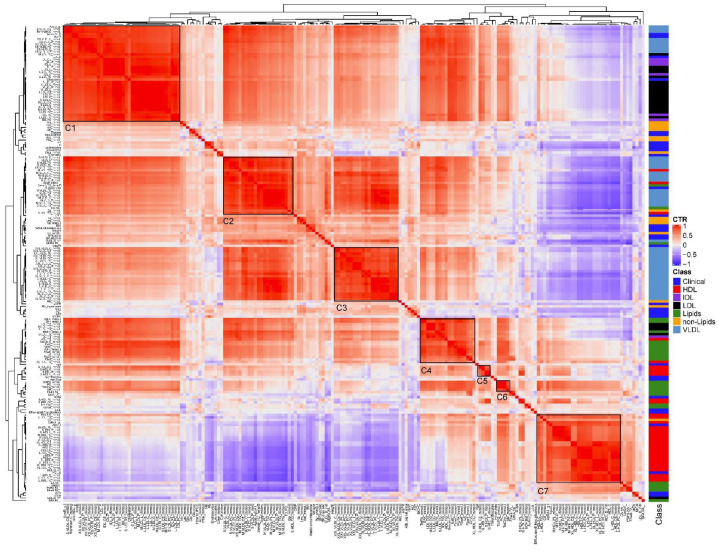
Correlation heatmap showing the correlations of clinical features and NMR parameters assessed in the control subjects. Red-blue heatmap reflects positive and negative correlations. Colors depicted in the vertical bar on the right side of the heatmap reflect subgroups of features: clinical findings and NMR features on VLDLs, LDLs, IDLs, HDLs, lipids in total plasma, and non-lipids. Cluster C1 encompasses apoB, non-HDL-cholesterol, LDL-cholesterol, and total cholesterol, as well as particle numbers, lipids, total, unesterified, and esterified cholesterol, and phospholipids of very small VLDL, IDL, small, medium, and large LDL. The cluster is loosely connected with glucose and HbA1c, as well as alanine, glutamine, phenylalanine, acetate, and lactate. Cluster C2 encompasses features of small and medium VLDL, triglycerides in total serum, very small, small, and medium VLDL, as well as small HDL and glycoprotein acetyls. This cluster is loosely connected with branched-chain amino acids, tyrosine, BMI, waist circumference, blood pressure, and CRP. Cluster C3 encompasses measures of large, very large, and extremely large VLDL. It is positively correlated with cluster C2 and, more loosely, with estimated glomerular filtration rate, histidine, albumin, and activities of liver enzymes. Cluster C4 contains triglycerides in total IDL, LDL, and HDL, as well as in small, medium, large LDL, very large HDL, and several features related to fatty acids, including total fatty acids, saturated fatty acids, docosahexaenoic acid, linoleic acid, MUFAs, PUFAs, omega-3 and omega-6 fatty acids. Moreover, it includes total, esterified, and unesterified cholesterol in plasma. Cluster C4 is connected with two smaller clusters, C5 and C6, and is loosely connected with age. Cluster C5 contains particle numbers, total and esterified cholesterol, and lipids of small HDL. Cluster C6 contains total phosphoglyceride, phosphatidylcholine, total cholesterol, and sphingomyelins. Cluster C7 encompasses apoA-I, citrate, and triglycerides in large HDL, lipids, phospholipids, total, unesterified, and esterified cholesterol, particle numbers of medium, large, and very large HDL, as well as total cholesterol in total HDL. It is loosely connected with unsaturated fatty acids, several fatty acid ratios, and cholesterol efflux capacity. Cluster C7 is inversely correlated with clusters C1, C2, and C3, as well as several features of cluster C4. Correlations with features of clusters C5 and C6 are mostly positive.

**Figure 3 biomolecules-13-00497-f003:**
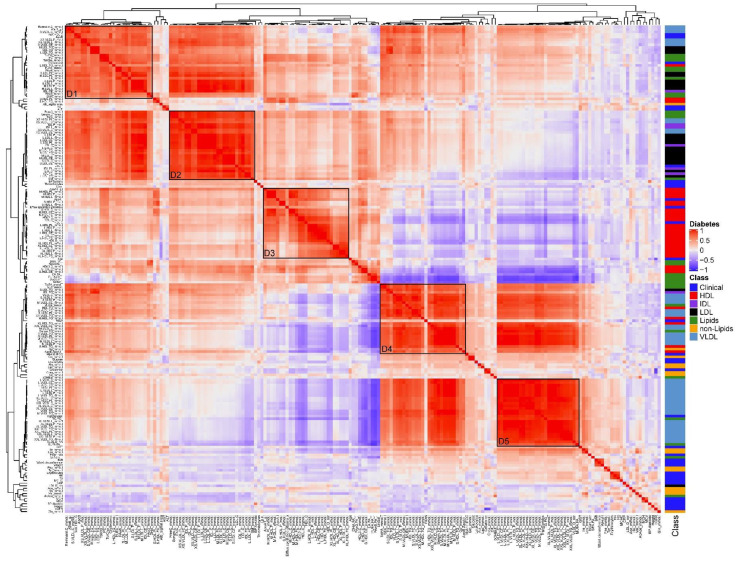
Correlation heatmap showing the correlations of clinical features and NMR parameters assessed in the patients with diabetes. Red-blue heatmap reflects positive and negative correlations. Colors depicted in the vertical bar at the right side of the heatmap reflect subgroups of features: clinical findings and NMR features on VLDLs, LDLs, IDLs, HDLs, lipids in total plasma, and non-lipids. Cluster D1 resembles non-diabetic subjects’ cluster C1 (Figure 2) through the presence of apoB, remnant cholesterol, total cholesterol, and non-HDL cholesterol, and cluster C4 through the presence of triglycerides in total, medium, large LDL, and large HDL. It also contains particle numbers, total lipids and phospholipids of medium and large LDL, omega-3 and omega-6 fatty acids, phosphatidylcholines, docosahexaenoic acid, PUFAs, and linoleic acid in plasma as well as total phosphoglycerides, total cholesterol, and cholesteryl esters in small VLDL and total lipids in very small VLDL. It is loosely connected with phospholipids and unesterified cholesterol of small HDL, as well as albumin and the activities of transaminases. Cluster D2 also resembles cluster C1 by the assembly of particle numbers, total, free, and esterified cholesterol, phospholipids of very small VLDL, IDL, large LDL, total lipids in IDL, large LDL, and cholesteryl esters in small and medium LDL and very small VLDL, as well as free cholesterol in small, medium and large LDL, total IDL, and very small VLDL. It is loosely connected with sphingomyelins of plasma, systolic blood pressure, and thrombocytes. Cluster D3 contains several parameters related to HDL, namely HDL-cholesterol and apoA-I, particle numbers, lipids, phospholipids, total, unesterified, and esterified cholesterol of medium, large, and very large HDL, particle number and total lipids of small HDL, as well as cholesterol efflux capacity. It resembles cluster C7. It is connected with clusters D1 and D2 by mostly positive correlations. However, most parameters of cluster D3 are inversely correlated with features of clusters D4 and D5. It is loosely connected to age, total cholesterol in HDL3, total and esterified cholesterol in small HDL, unsaturated fatty acids, and several ratios of fatty acids. Cluster D4 consists of measures of mostly small and medium VLDL and triglycerides in total serum (measured by NMR), very small and small VLDL, IDL, small LDL, and total, small, medium, and very large HDL. It is loosely connected with measures of glycemic control, alanine, lactate, citrate, leucocytes, and 3-hydroxybutyrate. Overall, cluster D4 shares similarities with clusters C2 and C4. Similar to cluster C3, cluster D5 encompasses measures of large, very large, and extremely large VLDL and triglycerides in serum measured by the clinical laboratory. The cluster is loosely connected with leucine, isoleucine, phenylalanine, and tyrosine, as well as measures of overweight and obesity, Gamma-GT, and CRP.

**Figure 4 biomolecules-13-00497-f004:**
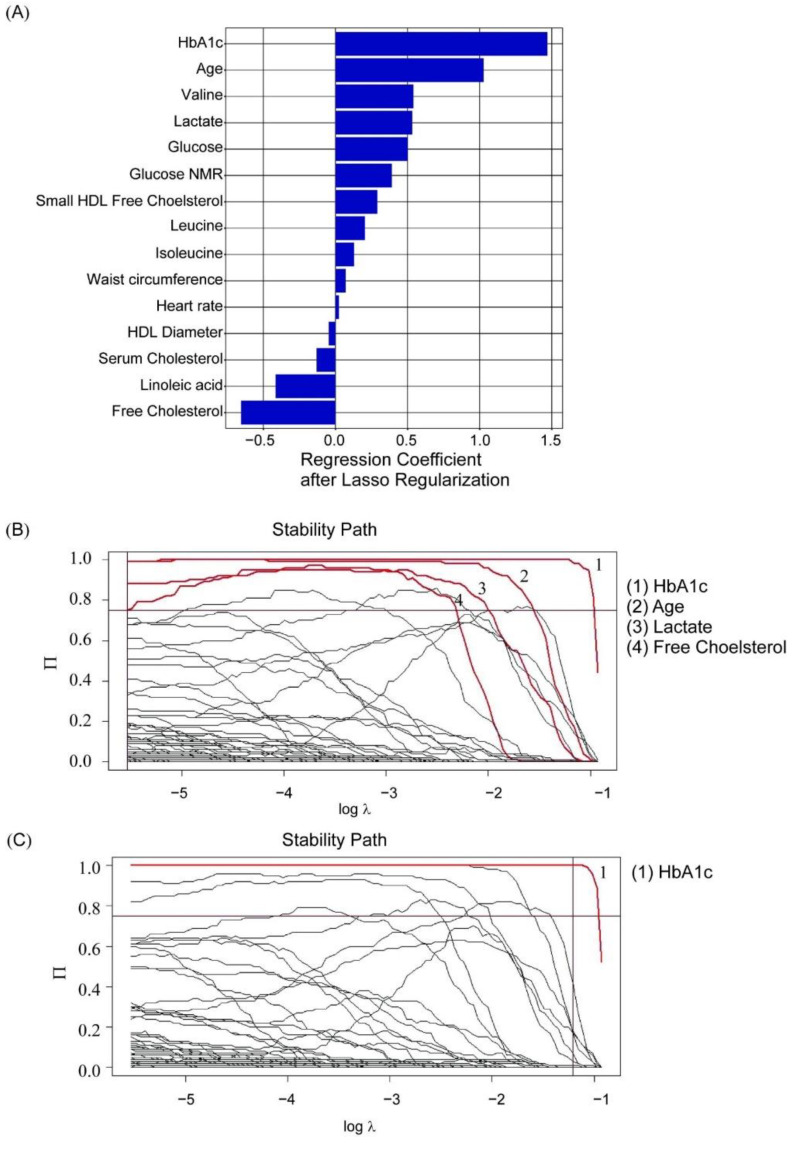
Elastic net regularized regression analyses on the association of variables with the presence of diabetes. (**A**) Bar plot showing regression coefficients for variables significantly associated with diabetes upon elastic-net analysis with lasso regression. All non-zero-coefficients after shrinking with lasso are depicted. The size of each bar reflects the strength of its association. The stability path of the regressions was analyzed either with the family-wise error rate (**B**) or the comparison-wise error rate (**C**). For regularization, the lasso regression method was used. Each line shows coefficients for one variable for different lambdas. The higher the lambda (x-axis), the more the coefficients are shrunk toward zero by regularization. The higher up on the y-axis, the higher the probability that a parameter is a non-zero parameter (it cannot be shrunk to zero). Red lines reflect parameters that are not shrunk to zero by the regression and are therefore stably associated with diabetes in the different models, namely HbA1c (both models B and C), age, lactate, and free cholesterol (model B only).

**Figure 5 biomolecules-13-00497-f005:**
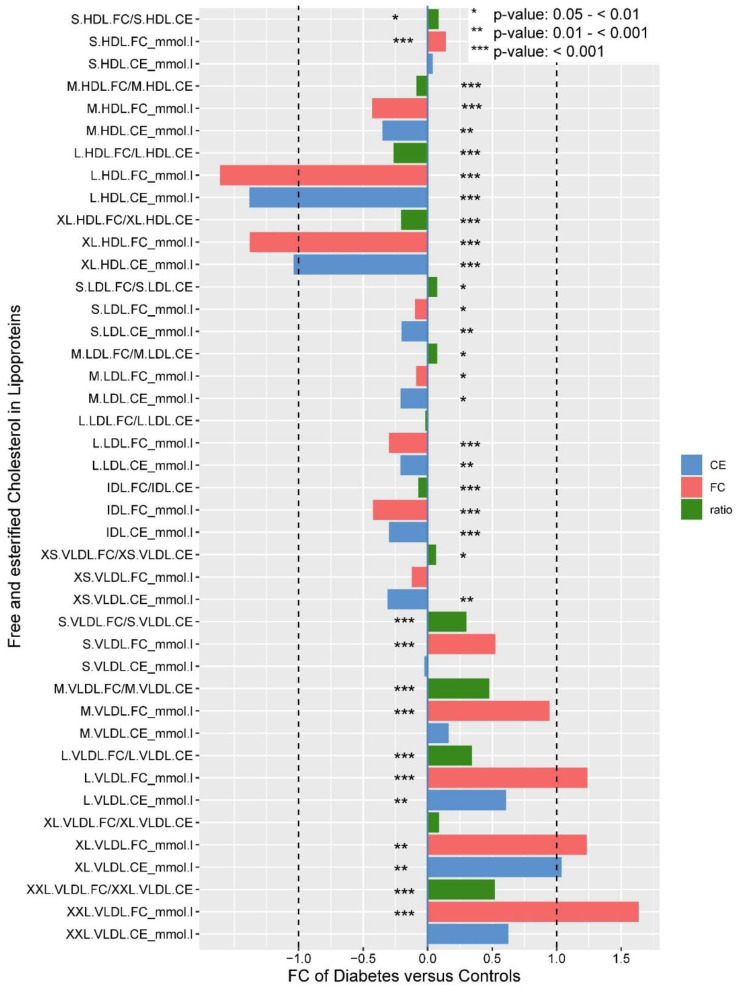
Bar plot showing associations of diabetes with free cholesterol (FC), cholesterol ester (CE), and the ratio of free cholesterol to total cholesterol in lipoprotein subclasses. Data are presented as multiples of the medians. Asterisks refer to the level of statistical significance as assessed by the Wilcoxon Test, adjusted by the Benjamin–Hochberg procedure: * *p* < 0.05; ** *p* < 0.01; *** *p* < 0.001.

**Figure 6 biomolecules-13-00497-f006:**
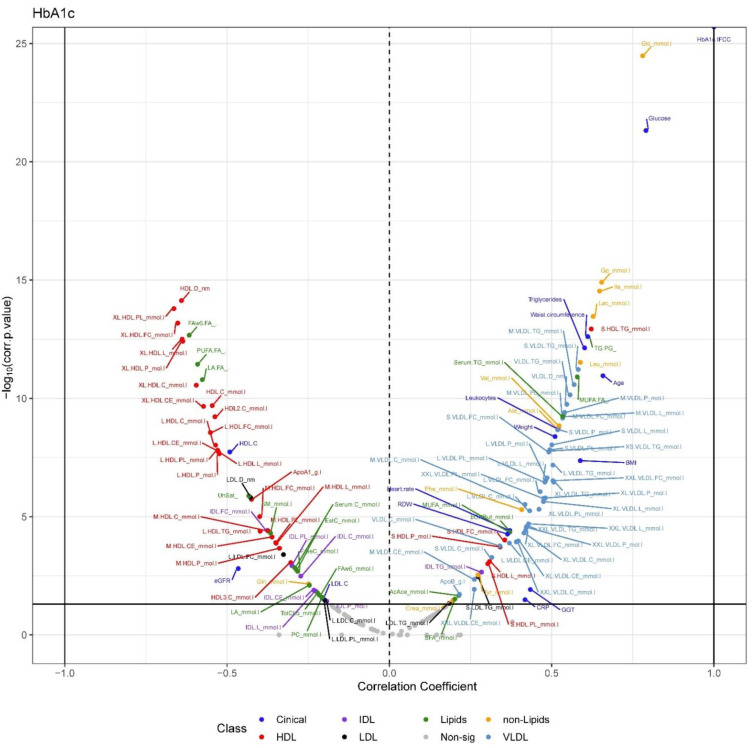
Volcano plot of correlations between HbA1c and all other parameters in the combined cohort of diabetic patients and non-diabetic controls. The x-axis reflects the coefficients of correlation. The y-axis depicts the level of statistical significance. The higher the location of a parameter in the graph, the higher the statistical significance of its correlation with HbA1c. The horizontal line reflects the threshold for statistical significance without adjustment for multiple testing. Only parameters with significant correlations are labeled with names. For abbreviations, see the list of abbreviations in Appendix A.

**Figure 7 biomolecules-13-00497-f007:**
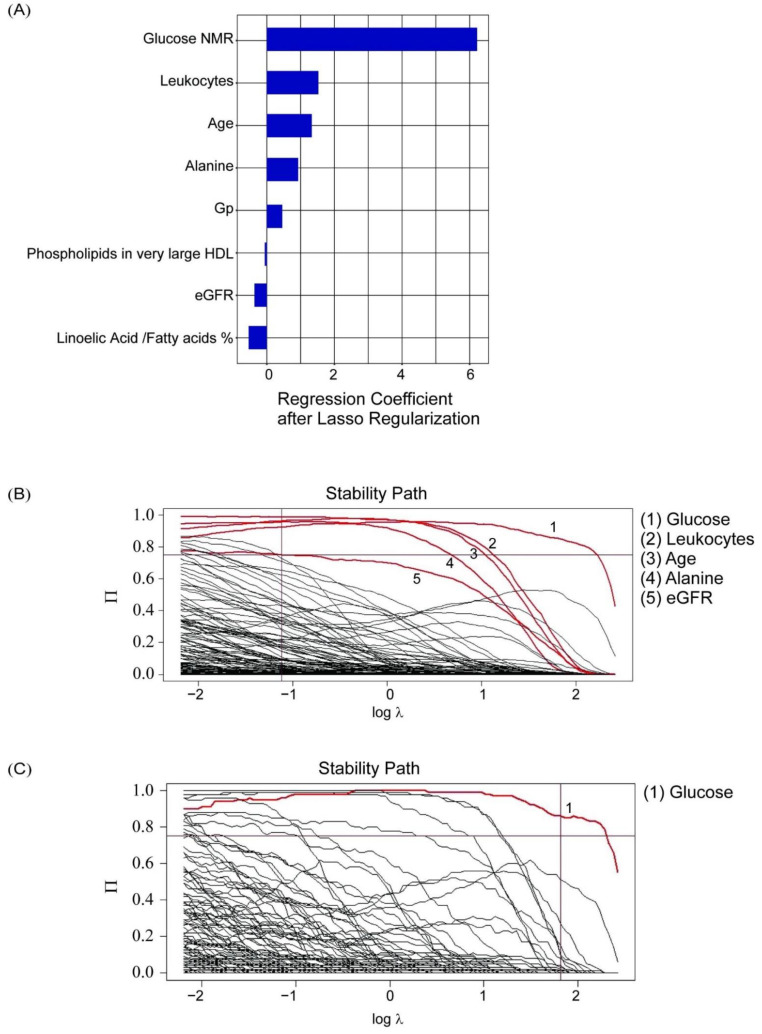
Elastic net regularized regression analyses on the association of variables with HbA1c. (**A**) Bar plot showing regression coefficients for variables significantly correlated with HbA1c upon elastic net analysis with lasso regression. All non-zero coefficients after shrinking with lasso are depicted. The size of each bar reflects the strength of its correlation. For regularization, the lasso regression method was used. The stability paths of the regressions were analyzed either with the (**B**) comparison-wise error rate or (**C**) family-wise error rate. Each line shows coefficients for one variable for different lambdas. The higher the lambda (x-axis), the more the coefficients are shrunk towards zero by regularization. The higher up on the y-axis, the higher the probability that a parameter is a non-zero parameter (it cannot be shrunk to zero). Red lines reflect parameters not shrunk to zero by the regression and are therefore stably associated with diabetes in the different models, namely glucose (both models B and C), leukocytes, age, alanine, and eGFR (model B only).

**Figure 8 biomolecules-13-00497-f008:**
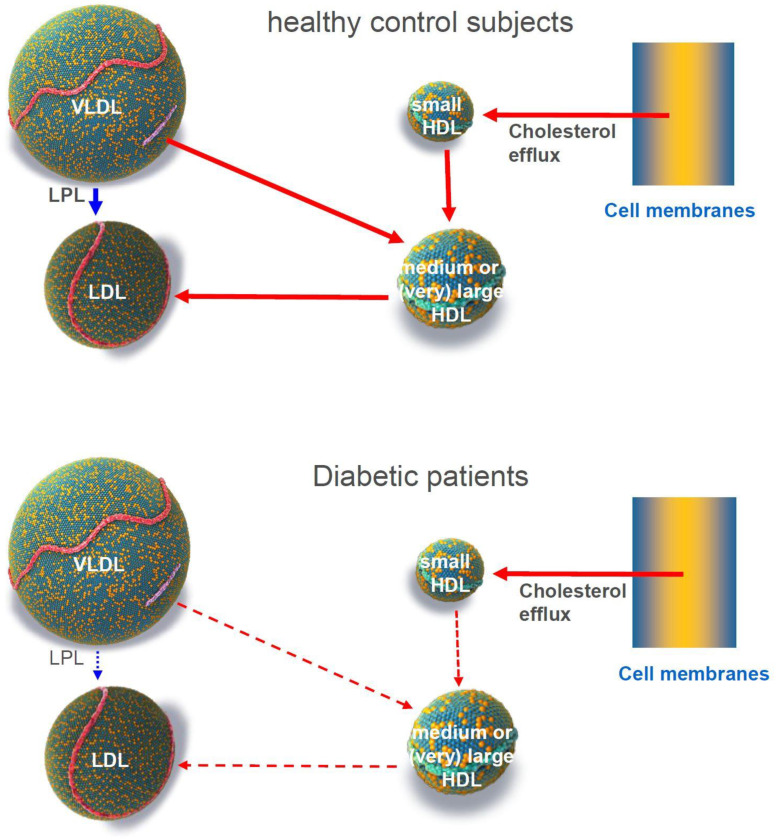
Illustration of the major findings and a model to explain the disbalanced distribution of free cholesterol among lipoproteins in diabetes mellitus.

**Table 1 biomolecules-13-00497-t001:** Demographic, anthropometric, and clinical characteristics.

Variable	Controls (N = 51)	T2DM Patients (N = 66)	Adjusted *p*-Value
Age (years)	39 (22–72)	61 (43–74)	**9.26 × 10^−13^**
Male (%)	35.3%	74.2%	**3.90 × 10^−9^**
CHD known	0	25.7%	
BMI (kg/m^2^)	23.31 (16.54–44.22)	30.74 (19.7–45.16)	**8.29 × 10^−11^**
Waist circumference (cm)	83.5 (65.6–127.6)	112.25 (78–141.8)	**1.57 × 10^−12^**
Smoking (%)	9.8%	18.2%	**2.17 × 10^−2^**
systolic BP (mmHg)	120 (98–158)	136 (95–183)	**2.57 × 10^−3^**
diastolic BP (mmHg)	80 (60–105)	85.5 (59–110)	**9.56 × 10^−3^**
Heart.rate (bpm)	68 (44–93)	76.5 (58–102)	**3.56 × 10^−5^**
Hemoglobin (g/L)	141 (115–168)	146.5 (100–189)	**3.42 × 10^−2^**
Hematocrit (L/L)	0.43 (0.36–0.48)	0.44 (0.33–0.57)	**2.15 × 10^−2^**
Erythrocytes (T/L)	4.61 (3.73–5.68)	4.88 (3.88–6.01)	**2.82 × 10^−2^**
Leukocytes (G/L)	5.5 (2.91–12.2)	7.42 (3.29–14.7)	**8.44 × 10^−7^**
Thrombocytes (G/L)	247 (135–342)	234 (46–498)	4.27 × 10^−1^
CRP (mg/L)	0.5 (0.21–9.4)	1.75 (0.21–40.6)	**4.07 × 10^−6^**
ALT (U/L)	18 (9–59)	26 (10–158)	**1.11 × 10^−3^**
AST (U/L)	22 (11–50)	24 (14–84)	1.62 × 10^−1^
GGT (U/L)	15 (3–80)	31.5 (11–216)	**7.02 × 10^−8^**
Creatinine (µmol/L)	72 (51–114)	82 (48–273)	**3.28 × 10^−4^**
eGFR (ml/min)	97 (69–124)	85 (20–109)	**8.22 × 10^−6^**
Glucose (mmol/L)	4.8 (3.8–5.9)	8.05 (3.8–20.4)	**1.10 × 10^−15^**
HbA1c.IFCC (nmol/mol)	35 (27–45)	55 (32–98)	**8.23 × 10^−16^**
Cholesterol (mmol/L)	4.7 (2.9–7.7)	4.1 (2.3–8.1)	**5.95 × 10^−5^**
Triglycerides (mmol/L)	0.79 (0.27–3.89)	1.665 (0.54–9.98)	**3.80 × 10^−9^**
HDL-C (mmol/L)	1.77 (0.8–2.68)	1.145 (0.58–1.98)	**3.85 × 10^−9^**
Non-HDL.C (mmol/L)	2.9 (1.4–6.5)	2.7 (1–7.3)	2.96 × 10^−1^
LDL-C (mmol/L)	2.5 (1–5.4)	1.95 (0.08–3.7)	**1.19 × 10^−4^**
Lp(a) (mg/dL)	101 (14.14–666)	93.5 (14.14–1816)	6.50 × 10^−1^
ApoAI (g/L)	1.62 (1.05–2.18)	1.37 (0.96–1.96)	**3.38 × 10^−5^**
ApoB (g/L)	0.83 (0.43–1.67)	0.8 (0.45–1.57)	8.51 × 10^−1^
Statin (%)	0	72.6	
Insulin (%)	0	62.9	
Metformin (%)	0	75.8	
GLP1-related drugs (%)	0	51.6	
Sulfonylurea (%)	0	12.9	
SGLT2 Inhibitor (%)	0	17.7	

Data are presented as medians (and ranges) or prevalences (in %). *t*-test and Wilcoxon test were used for parameters with and without Gaussian frequency distribution, respectively.

## Data Availability

The data presented in this study are available on request from the corresponding author.

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
