# Peer review of "Altered Distribution of Unesterified Cholesterol among Lipoprotein Subfractions of Patients with Diabetes Mellitus Type 2"

_biomolecules, 2023, doi:10.3390/biom13030497_

Round 1
Reviewer 1 Report
The presence of diabetes independently associated with lower concentrations of free cholesterol in plasma but higher concentrations of free cholesterol in small HDL. Compared to plasmas of non-diabetic controls, plasmas of diabetic subjects contained lower absolute and relative concentrations of free cholesterol in all LDL and HDL subclasses except small HDL but higher absolute and relative concentrations of free cholesterol in all VLDL subclasses (except very small VLDL). These disbalances may reflect disturbances in the transfer of free cholesterol from VLDL to HDL during lipolysis and in the transfer of cell-derived cholesterol from small HDL via larger HDL to LDL.
It is interesting and meaningful discovery to test the higher concentrations of free cholesterol in small HDL. However, there still have some other issues need to clear.
1. The hyperlipidemia is always companied by diabetes, which is metabolic syndrome.
2. Glycosylated hemoglobin or glycated hemoglobin(GHb) is a typical marker of diabetic subjects. Is there any link of AGEs with free cholesterol in small HDL? Please refer the analyze in this article (Dietary polyphenols: regulate the advanced glycation end products (AGEs)-RAGE axis and the microbiota-gut-brain axis to prevent neurodegenerative diseases. Critical Reviews in Food Science and Nutrition. Doi: 10.1080/10408398.2022.2076064.)
3. The Lipid type and quantity is critical to cholesterol content in plasma.
4. The effect of Insulin resistant should be considered.
5. The typical NMR information or map about cholesterol should be supplement.
6. The diet factor should be analyzed (Whole grain benefit: oat β-glucan and phenolic compounds synergistically regulates hyperlipidemia via gut microbiota in high-fat-diet mice. Food & Function, 2022, 13, 12686-12696. Doi: 10.1039/d2fo01746f.).
Author Response
Rebuttal to Reviewer 1
The presence of diabetes independently associated with lower concentrations of free cholesterol in plasma but higher concentrations of free cholesterol in small HDL. Compared to plasmas of non-diabetic controls, plasmas of diabetic subjects contained lower absolute and relative concentrations of free cholesterol in all LDL and HDL subclasses except small HDL but higher absolute and relative concentrations of free cholesterol in all VLDL subclasses (except very small VLDL). These disbalances may reflect disturbances in the transfer of free cholesterol from VLDL to HDL during lipolysis and in the transfer of cell-derived cholesterol from small HDL via larger HDL to LDL. It is interesting and meaningful discovery to test the higher concentrations of free cholesterol in small HDL. However, there still have some other issues need to clear.
We thank the reviewer for his or her time and effort as well as helpful comments to improve the manuscript
1. The hyperlipidemia is always companied by diabetes, which is metabolic syndrome.
The reviewer is principally right. In fact in our cohort the diabetic patients, the featrues of metabolic syndrome were more prevalent as they had higher waist circumference, blood pressure, higher triglycerides and lower HDL-C. These are important confounders that were considered in our multivariable analyses. However, except waist circumference (and HbA1c as the measure of glycemia) none remained statistical significant. And the association of free cholesterol both in plasma and HDL was stronger than that of waist circumference. We acknowledge this in the limitation section of our discussion (line 365)
2. Glycosylated hemoglobin or glycated hemoglobin(GHb) is a typical marker of diabetic subjects. Is there any link of AGEs with free cholesterol in small HDL? Please refer the analyze in this article (Dietary polyphenols: regulate the advanced glycation end products (AGEs)-RAGE axis and the microbiota-gut-brain axis to prevent neurodegenerative diseases. Critical Reviews in Food Science and Nutrition. Doi: 10.1080/10408398.2022.2076064.)
Features of small HDL including free cholesterol in small HDL are positively correlated with HbA1c whereas features if medium, large and very large HDL have inverse correlations. However, upon multivariate analysis none of the small HDL features remains correlated with HbA1c. (see red items in figure 7). We describe this also in the text of the original and revised version of the manuscript (lines 275-277 of the tracked version). We have no data on microbiota or neurodegenerative diseases so that a citation of the interesting paper does not appear justified
3. The Lipid type and quantity is critical to cholesterol content in plasma.
We do not understand the comment. Does the reviewer mean the correlation of cholesterol with other classes of lipids. This is generally correct and illustrated by the heat maps. Or does the reviewer mean the heterogeneity of fatty acids in cholesteryl esters? This is not relevant for free cholesterol because it does not contain any fatty acids. It is a clearly defined molecule. Or does the reviewer sterols other than cholesterol, for example oxysterols or phytosterols? We have no data on them. However, their concentrations are several orders of magnitude lower than that of cholesterol
4. The effect of Insulin resistant should be considered.
In the discussion, we added insulin resistance as the cause of low lipoprotein lipase activity and hence decreased transfer of free cholesterol from VLDL to HDL (line 340 of the tracked version)
5. The typical NMR information or map about cholesterol should be supplement.
We do not understand the comment. Does the reviewer want us to show an NMR spectrum?
6. The diet factor should be analyzed (Whole grain benefit: oat β-glucan and phenolic compounds synergistically regulates hyperlipidemia via gut microbiota in high-fat-diet mice. Food & Function, 2022, 13, 12686-12696. Doi: 10.1039/d2fo01746f.).
Unfortunately we have no data on the dietary behavior of the probands. We admit that this will be interesting but require dietary intervention studies rather than analyses of correlations between observational data obtained by food questionnaires. In the revised version, we add an according comment in the limitation section of the discussion (line 365 of the tracked version). We don’t cite the interesting reference because this study investigated other endpoints than we in our study .
Reviewer 2 Report
Kolb and co-authors have written an interesting and well-conceived manuscript on using biomarkers as disease predictors in patients at high risk of developing type 2 diabetes and as stratification indices of diabetic patients according to risk factors. The authors analyzed more than 150 metabolic parameters in serum and plasma samples of diabetic patients and healthy subjects using nuclear magnetic resonance (NMR) spectrometry.
Here are my considerations for the authors:
- Have the different parameters in the two populations been evaluated, distinguishing between men and women? have differences emerged?
- The population of healthy subjects has higher LDL-cholesterol values than the diabetic population, and both are in normal ranges. Can you confirm this? Is there a correlation with your results?
- NMR allows using of serum or plasma. Why did you analyze plasma?
- It would be interesting to characterize diabetic patients according to their concomitant pathologies.
- Why more than 70% of diabetic patients take a statin?
- It would be appropriate to explain the regularization of Lasso concisely.
- Since Figures 2 and 3 are very complex, I suggest adding extra data in the caption or text to interpret them better.
- If possible, a representative figure of the results obtained and the hypothesis made by the authors regarding the application of these results in research and clinical practice.
- I suggest a careful reading of the text for the presence of some typos
Author Response
Rebuttal to Reviewer 2
Kolb and co-authors have written an interesting and well-conceived manuscript on using biomarkers as disease predictors in patients at high risk of developing type 2 diabetes and as stratification indices of diabetic patients according to risk factors. The authors analyzed more than 150 metabolic parameters in serum and plasma samples of diabetic patients and healthy subjects using nuclear magnetic resonance (NMR) spectrometry.
We thank the reviewer for his or her time and effort as well as helpful comments to improve the manuscript
Here are my considerations for the authors:
- Have the different parameters in the two populations been evaluated, distinguishing between men and women? have differences emerged?
In fact males were overrepresented by factor 2 in the group of diabetic patients. We included gender in the multivariate analysis and it did not emerge as an independent determinant of either diabetes or HbA1c. Nevertheless we admit the important role as a confounder in the revised limitation part of the discussion (lines 368 to 369 of the tracked version)
- The population of healthy subjects has higher LDL-cholesterol values than the diabetic population, and both are in normal ranges. Can you confirm this? Is there a correlation with your results?
The observation is correct. However, in general patients with type 2 diabetes have no higher levels of LDL-cholesterol than non-diabetic subjects. In our cohort it is even lower, because of the wide spread use of statins among these patients. (see your comment 5). LDL-C levels were rather low in our control subjects probably because recruitment via advertisements generated a bias towards subjects with a healthy life style. Moreover, LDL-C levels increase by age and with a mean age of 39 years, our control subjects were rather young. LDL-C was incorporated in our multivariate analyses and did not emerge as an independent confounder.
- NMR allows using of serum or plasma. Why did you analyze plasma?
We only had EDTA plasma available. The protocol of Nightingale®) allows the analysis of both serum and plasma for NMR spectroscopy.
- It would be interesting to characterize diabetic patients according to their concomitant pathologies.
Of our diabetic subjects, 17 were known to have coronary heart disease. We add this information to table 1.
- Why more than 70% of diabetic patients take a statin?
At the time of recruitment, guidelines recommend to target LDL-C levels < 1.8 mmol/L in patients with diabetes and CHD and to LDL-C levels < 2.6 mmol/L in all other diabetic patients. Thus, the high prevalence of statin treatment is not surprising but reflects good translation of European prevention guidelines into clinical practice at the recruiting center.
- It would be appropriate to explain the regularization of Lasso concisely.
We extended the description of the Lasso regularization (lines 109-111 of the tracked version)
- Since Figures 2 and 3 are very complex, I suggest adding extra data in the caption or text to interpret them better.
We added an explanatory text to the legends of the two figures (lines 185-200 and lines 207 to 225 of the tracked version)
- If possible, a representative figure of the results obtained and the hypothesis made by the authors regarding the application of these results in research and clinical practice.
We conclude the discussion section with a novel figure 10 as the graphical abstract and a sentence on future research goals (lines 377 to 380 of the tracked version).
- I suggest a careful reading of the text for the presence of some typos
Detailed proof reading of the manuscript unraveled some typos and ambiguities that were corrected

Round 2
Reviewer 1 Report
The presence of diabetes independently associated with lower concentrations of free cholesterol in plasma but higher concentrations of free cholesterol in small HDL. Compared to plasmas of non-diabetic controls, plasmas of diabetic subjects contained lower absolute and relative concentrations of free cholesterol in all LDL and HDL subclasses except small HDL but higher absolute and relative concentrations of free cholesterol in all VLDL subclasses (except very small VLDL). These disbalances may reflect disturbances in the transfer of free cholesterol from VLDL to HDL during lipolysis and in the transfer of cell-derived cholesterol from small HDL via larger HDL to LDL.
It is interesting and meaningful discovery to test the higher concentrations of free cholesterol in small HDL. However, the author still has some other issues need to clear. The response to reviewer’s comments still need to improve.
1. The hyperlipidemia is always companied by diabetes, which is metabolic syndrome.
The serum lipid type is critical to cholesterol content in plasma associated with diebates (Whole grain benefit: oat β-glucan and phenolic compounds synergistically regulates hyperlipidemia via gut microbiota in high-fat-diet mice. Food & Function, 2022, 13(24), 12686-12696. Doi: 10.1039/d2fo01746f.).
2. The effect of correlation of glycolipid metabolism with diets should be considered for the biomarkers(Oat phenolic compounds regulate metabolic syndrome in high fat diet-fed mice via gut microbiota. Food Bioscience. 50(2022)101946. Doi: 10.1016/j.fbio.2022.101946 ).
3. The relationship of HbA1c with AGEs should be discussed(Dietary polyphenols: regulate the advanced glycation end products (AGEs)-RAGE axis and the microbiota-gut-brain axis to prevent neurodegenerative diseases. Critical Reviews in Food Science and Nutrition. Doi: 10.1080/10408398.2022.2076064.).
4. The language expression should be improved.
Author Response
Rebuttal to Reviewer 1
The presence of diabetes independently associated with lower concentrations of free cholesterol in plasma but higher concentrations of free cholesterol in small HDL. Compared to plasmas of non-diabetic controls, plasmas of diabetic subjects contained lower absolute and relative concentrations of free cholesterol in all LDL and HDL subclasses except small HDL but higher absolute and relative concentrations of free cholesterol in all VLDL subclasses (except very small VLDL). These disbalances may reflect disturbances in the transfer of free cholesterol from VLDL to HDL during lipolysis and in the transfer of cell-derived cholesterol from small HDL via larger HDL to LDL. It is interesting and meaningful discovery to test the higher concentrations of free cholesterol in small HDL. However, the author still has some other issues need to clear. The response to reviewer’s comments still need to improve.
Reply: We thank the reviewer for his or her time and effort in reviewing our manuscript a second time. With respect to free cholesterol in small HDL, we like to draw the reviewer’s attention to reference 15 of our manuscript. In this analysis of UK-Biobank data, free cholesterol in HDL was associated with incident diabetes.
The hyperlipidemia is always companied by diabetes, which is metabolic syndrome. The serum lipid type is critical to cholesterol content in plasma associated with diebates (Whole grain benefit: oat β-glucan and phenolic compounds synergistically regulates hyperlipidemia via gut microbiota in high-fat-diet mice. Food & Function, 2022, 13(24), 12686-12696. Doi: 10.1039/d2fo01746f.).
Reply: this comment repeats and combines the comments 1, 3 and 6 of this reviewer on our first submission. We addressed the comment on metabolic syndrome and diet in the according rebuttal and first revision by adding according comments in the limitation section of the discussion. Without any further and more specific criticism, we do not understand what the reviewer expects us to do. Does he or she simply wants us to cite the indicated publication. This will be artificial, because the content of this publication has nothing to do with our work.
The effect of correlation of glycolipid metabolism with diets should be considered for the biomarkers(Oat phenolic compounds regulate metabolic syndrome in high fat diet-fed mice via gut microbiota. Food Bioscience. 50(2022)101946. Doi: 10.1016/j.fbio.2022.101946 ).
Neither do we have any data on diet or gut microbiota nor does the NMR analysis provide any data on glycolipids. Insofar we do not see any context in which we can discuss these aspects and cite the paper indicated by the reviewer.
The relationship of HbA1c with AGEs should be discussed(Dietary polyphenols: regulate the advanced glycation end products (AGEs)-RAGE axis and the microbiota-gut-brain axis to prevent neurodegenerative diseases. Critical Reviews in Food Science and Nutrition. Doi: 10.1080/10408398.2022.2076064.).
Reply: This comment repeats the previous comment 2 of this reviewer on our original submission. As discussed in our previus rebuttal, we have not presented any data on microbiota and neurodegeneration so that any discussion of these aspects and the citation of the indicated paper will be out of context.
The language expression should be improved.
Reply: The manuscript was proofread by all authors. The editors should decide if there is any need for further language improvement